# No Foundations without Foundations: Why semi-mechanistic models are essential for regulatory biology

## Abstract

Despite substantial efforts, deep learning has not yet delivered a transformative impact on elucidating regulatory biology, particularly in the realm of predicting gene expression profiles. Here, we argue that genuine "foundation models" of regulatory biology will remain out of reach unless guided by frameworks that integrate mechanistic insight with principled experimental design. We present one such ground-up, semi-mechanistic framework that unifies perturbation-based experimental designs across both *in vitro* and *in vivo* CRISPR screens, accounting for differentiating and non-differentiating cellular systems. By revealing previously unrecognised assumptions in published machine learning methods, our approach clarifies links with popular techniques such as variational autoencoders and structural causal models. In practice, this framework suggests a modified loss function that we demonstrate can improve predictive performance, and further suggests an error analysis that informs batching strategies. Ultimately, since cellular regulation emerges from innumerable interactions amongst largely uncharted molecular components, we contend that systems-level understanding cannot be achieved through structural biology alone. Instead, we argue that real progress will require a first-principles perspective on how experiments capture biological phenomena, how data are generated, and how these processes can be reflected in more faithful modelling architectures.

## 1 Introduction

Three main themes presently dominate machine learning (ML) research in biology: structural biology [see AlphaFold 2024], sequence modelling [including DNA (Avsec et al., 2021), RNA (Sumi et al., 2024), and proteins (Zhou et al., 2024)], and regulatory biology. Regulatory biology harbours a key unsolved problem: understanding the mapping between the manipulation of genes (e.g., knockout, inhibition, or overexpression) and a resulting complex downstream phenotype (e.g., proliferation, cytotoxicity, or extracellular matrix production) — a longstanding Grand Challenge known as the 'genotype–phenotype relationship' (Uhler, 2024). Understanding which gene manipulations lead to changes in phenotype that are considered beneficial is a fundamental task in drug discovery since it opens the possibility of seeking drugs that mimic those perturbations. Despite billions of dollars spent on drug development, success rates remain low, with the principal cause of failure being an absence of efficacy (meaning a drug fails to exert a beneficial effect) (Taylor-King et al., 2024) — in other words, *a failure to accurately predict the effect of a perturbation.*

Historically, biological assays were exclusively low throughput and collapsed high-dimensional regulatory states into a single value (e.g. a phenotypic measure). However, now we have the ability to generate large amounts of perturbation data suitable for ML through the use of pooled

CRISPR screens with single-cell readouts (Frangieh et al., 2021; Papalexi et al., 2021; Mimitou et al., 2019; Datlinger et al., 2017; Dixit et al., 2016) or arrayed screens (with appropriate automation). Other imaging-based readouts have also been scaled for genome-scale perturbations, for example, optical pooled screens (Gentili et al., 2024) and cell painting (Chandrasekaran et al., 2023). Recent computational models have explored the prediction of transcriptomic states for unseen perturbations — with the aim to understand biological pathways and improve downstream phenotype prediction (Roohani et al., 2022; Hetzel et al., 2022; Lotfollahi et al., 2019, 2021; Inecik et al., 2022).

Despite extensive research efforts, simple statistical methods continue to outperform deep learning in predicting transcriptomic profiles (Gaudelet et al., 2024; Ahlmann-Eltze et al., 2024; Wu et al., 2024; Bendidi et al., 2024; Wenteler et al., 2024). It is implausible that the underlying regulatory mechanisms are genuinely this trivial, so these shortfalls likely reflect two intertwined deficits: insufficient curated data and an overreliance on purely data-driven architectures. **We argue that the dream of "foundation models" in regulatory biology, those capable of robust and generalisable predictions, will remain elusive unless grounded in a biologically informed, semi-mechanistic framework.**

Building frameworks to model regulatory biology is a challenging task because of the complex nature of gene–gene interactions, e.g., physical protein–protein interactions, epistasis, and pleiotropy. Furthermore, even with the latest functional genomics techniques, it is not experimentally tractable to exhaustively screen all genes in isolation when using primary cells, and combinations of genes are not possible even when cell numbers are immaterial (for example, when using immortalized cell models) (Bertin et al., 2023). Finally, the standard CRISPR-Cas9 toolbox is constantly evolving, we can perform knockouts (Lara-Astiaso et al., 2023), but also activation (Norman et al., 2019) (via CRISPRa), interference (Tian et al., 2019) (via CRISPRi or CRISPR-Cas13), base editing, and prime editing (Przybyla and Gilbert, 2022). Foundation models typically draw upon data from a range of sources; when we consider the range of cell types, culture conditions, and emerging perturbation technologies available, we must develop sophisticated ways of describing experimental systems for integration purposes.

In this paper, we develop a semi-mechanistic mathematical model that captures interventions in pooled CRISPR screens with single-cell readouts, and show how this framework applies equally to other perturbation types and experimental designs (including both differentiating and non-differentiating cellular systems). This "ground-up" approach highlights subtle assumptions—often unvalidated—that underlie widely used methods, thus motivating generation of new datasets for rigorous testing. We also propose modifications to generic loss functions that incorporate key biological intuitions and demonstrate, on a published dataset, that such modifications achieve faster and more robust performance than standard alternatives. Our overarching position is that only by weaving mechanistic understanding with rigorous mathematical underpinnings can we scale foundation models to achieve the next generation of predictive, interpretable, and clinically valuable models in regulatory biology.

In Section 2, we provide a biologically-grounded mathematical model of an *in vitro* pooled CRISPR screen with single-cell readout, and show how this leads to different loss functions. In Section 3, we consider how single-cell technologies are views over a hidden cell state, which gives us insights into the relationship between batch effects and learned functions. In Section C, we discuss other experimental systems and in Section D, we show how the proposed mathematical framework connects many popular established ML models. In Section 4, we give a proof of principle demonstration of our approach using a neural ordinary differential equation (NODE) model, before providing a discussion in Section 5.

## 2 Modelling cell perturbation dynamics

For foundation models to achieve genuine out-of-distribution performance, we need to encode some conceptualization of how cells behave. Here, we describe a perturb-seq experiment and subsequently build a mathematical description to highlight the subtle assumptions made by other ML models.

### 2.1 Typical *in vitro* perturb-seq experiment description

Functional genomic screens typically first rely on a technology to manipulate the function or expression of genes followed by a downstream readout of cellular function. We focus this initial exposition

on a perturb-seq style system, i.e., a pooled CRISPR based screen with single-cell readout. However, this could easily apply to a phenotypic screen, an arrayed screen, etc.

In perturb-seq style screens, a large number of cells are simultaneously edited targeting a range of biological processes in a manner that allows for identification of the originating perturbation (Dixit et al., 2016; Datlinger et al., 2017). Perturbation technologies include knock outs (via CRISPR nuclease; CRISPRn), knock down (via CRISPR interference; CRISPRi), or overexpression (via CRISPR activation; CRISPRa) applied to a specified set of genes. Gene targeting is achieved through delivery of a CRISPR protein that will localise to a region of the genome via a single guide RNA (sgRNAs).

In some screens, cells are separated and treated with additional stimuli (Dräger et al., 2021); typically using cytokines chosen to induce a biological process of interest. We then wish to understand how this induced process is altered by the earlier genetic perturbation. Other stimuli also considered include small molecule drug screens (Srivatsan et al., 2020), or even co-culture (with a second cell type) as a new "media" condition (Frangieh et al., 2021).

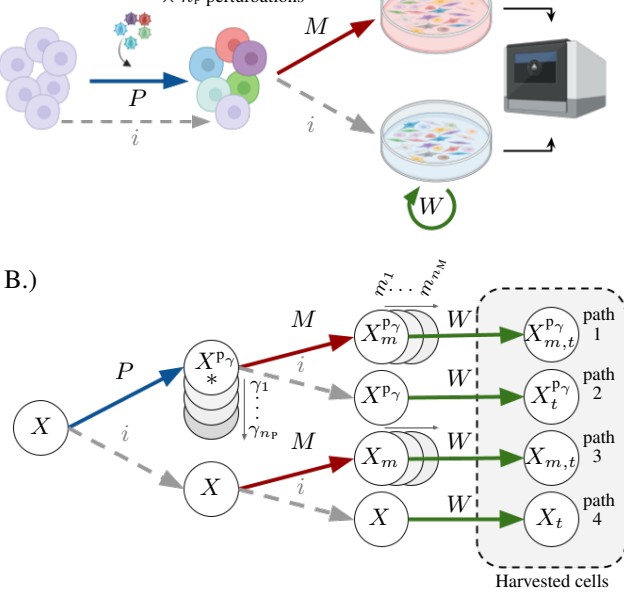

Figure 1: Illustration of abstracted phases within a perturb-seq experiment: application of a genetic perturbation, $P$; a change in a media condition, $M$; and the culturing of cells over time, $W$. In panel (A.) we provide a typical wet lab illustration, and in (B.) a branching process illustration with $(n_\mathrm{P} + 1)(n_\mathrm{M} + 1)$ total unique branches.

After some period of time whereby cells are cultured and maintained, cells are harvested and sequenced to understand how the perturbation and application of media leads to dysregulation of chromatin accessibility (Liscovitch-Brauer et al., 2021; Rubin et al., 2019; Pierce et al., 2021), the transcriptome (Lara-Astiaso et al., 2023), or select members of the proteome via oligonucleotide-tagged antibodies (Frangieh et al., 2021). See Figure 1A for a cartoon of this experiment. Resources now exist performing meta-analysis across such experiments and provide easy access to standardised data (Peidli et al., 2024).

## 2.2 Mathematical description

We abstract the *in vitro* perturb-seq experiment in Figure 1A to a sequence of three actions being performed: i.) the *instantaneous* application of a functional genomic perturbation; ii.) the *instantaneous* change of a cellular media condition; and iii.) a waiting period whereby cells are cultured and free to respond to changes induced by (i.) or (ii.). **Crucially, this order is important as these actions are not commutative.** Consider the transforming growth factor beta (TGF$\beta$) signalling pathway induced by specific molecules called TGF$\beta$ cytokines. One example of such molecules is *TGFB1*, which can be applied to cells through culture media. If the *TGFBR1* co-receptor was knocked out before applying *TGFB1*, the cascade cannot start. However, knocking out *TGFBR1* after stimulation with *TGFB1* would have no effect because the cascade has already begun – clearly the order of operations matters! We do not yet introduce the act of measuring cell state, introduced in Section 3.

We want to describe the internal state of a cell. In the absence of a highly technical mathematical construction, we describe a cell at rest (a 'control' cell) by random variable $X$ (in some undefined space $\mathcal{X}$ of random variables). Without being too specific, this cell could be in minimum essential media to maintain cell growth (i.e., amino acids, carbohydrates, vitamins, minerals, growth factors,

143 hormones, and gases); we refer to this as the *baseline media* condition. We annotate a gene $\gamma$ by
144 perturbation status $\mathrm{p}_\gamma$ driven by one of the aforementioned CRISPR technologies: $\mathrm{p}_\gamma = \times$ for
145 CRISPRn; $\mathrm{p}_\gamma = \downarrow$ for CRISPRi; $\mathrm{p}_\gamma = \uparrow$ for CRISPRa; and for completeness $\mathrm{p}_\gamma = \cdot$ for unperturbed.

146 Cells are then targeted and modified by CRISPR with associated apparatus, and the gene targeted by
147 the relevant sgRNA is perturbed. We represent this action by a function $P$ that applies $\mathrm{p}_\gamma$ to $X$, we
148 write

$$P(X, \mathrm{p}_\gamma) = X^{\mathrm{p}_\gamma}. \tag{1}$$

149 Here, we present a few properties of $P$. We first note that one cannot repeatedly knock out the same
150 gene, therefore

$$P(P(X, \mathrm{p}_\gamma = \times), \mathrm{p}_\gamma = \times) = P(X, \mathrm{p}_\gamma = \times).$$

151 Second, in this "instantaneous" framework, we specify that genetic perturbations are commutative in
152 the case where perturbations occur at the same point of time

$$(X^{\mathrm{p}_\gamma})^{\mathrm{p}_\delta} = (X^{\mathrm{p}_\delta})^{\mathrm{p}_\gamma} = X^{\mathrm{p}_\gamma \mathrm{p}_\delta} \quad \text{for} \quad \gamma \neq \delta.$$

153 Since one cannot apply multiple CRISPRi or CRISPRa to the same gene, operations like
154 $P(P(X, \mathrm{p}_\gamma = \uparrow), \mathrm{p}_\gamma = \downarrow)$ are not well defined. We note that gene dosing effects can be achieved
155 through CRISPRi with semi-efficacious sgRNA (Jost et al., 2020), however we do not address
156 these niche experimental set ups at this point[1]. Finally, we note that we model the application of a
157 non-targeting CRISPR construct as the identity function $P(X, \cdot) \equiv i(X) = X$.

158 For simplicity of exposition, we do not distinguish between edited cells containing a non-targeting
159 sgRNA, and unedited cells. Non-targeting or "scrambled" controls are typically used in perturb-seq
160 experiments in place of untransfected cells as one may wish to discount any stress response induced
161 by introduction of sgRNA. These effects are believed to be less relevant for longer experiments. If
162 these effects are in fact material, then such non-targeting sgRNA infected cells could be reclassified
163 as a perturbed population in its own right — meaning that the mathematical model need not change.

164 For the next phase of the experiment, a change in the media is made[2]. We represent this as the
165 function $M$ which applies media $m$ to random variable $X$, we write

$$M(X, m) = X_m. \tag{2}$$

166 As before, if no media change is made, the identity function is used, $M(X, \cdot) \equiv i(X) = X$. Similar
167 to the use of non-targeting sgRNAs, chemical experiments often use dimethyl sulfoxide (DMSO) as a
168 sham addition of media, but we do not cover these effects here.

169 Finally, the cells are left for a $t$ units of time, and the cell state is modified by waiting function $W$,
170 thus

$$W(X, t) = X_t, \tag{3}$$

171 and $W(X, 0) = X_0 \equiv X$.

172 The whole *in vitro* perturb-seq experiment described in Section 2.1 can then be abstracted to become
173 the application of the function

$$F(X, \mathrm{p}_\gamma, m, t) := (W \circ M \circ P)(X, \mathrm{p}_\gamma, m, t) = W(M(P(X, \mathrm{p}_\gamma), m), t)$$
$$= [(X^{\mathrm{p}_\gamma})_m]_t = X^{\mathrm{p}_\gamma}_{m,t}. \tag{4}$$

174 Here, we will always assume that $F$ encodes this specific order of operations and we write
175 $[(X^{\mathrm{p}_\gamma})_m]_t = X^{\mathrm{p}_\gamma}_{m,t}$, i.e. $W \circ M \circ P$ is non-commutative.

176 To reiterate, for certain experiments the order of operations is crucial. For example, editing out genes
177 that prevent cellular differentiation would have no effect if the target cells have been exposed to

---

[1]Natural gene-dosing effects may also occur through the use of knockouts whereby mixes of functional and non-functional genes coexist within diploid or aneuploid cell models. However, robust quality control can remove or account for such effects.

[2]Typical media changes include the addition of small molecules and cytokines. From a ML perspective, small molecules can be represented via their structure, cytokines may be characterised by their amino acid sequence. Whilst cytokines are likely to be somewhat characterised by a receptor they interact with, small molecules may interact with a plethora of proteins (Gaudelet et al., 2021).

differentiation-inducing media prior to the perturbation. Similarly, if a toxic genetic perturbation is applied to a cell before being exposed to media, the effect would be the same regardless of media applied. Under these circumstances, a different order of operations would lead to a different outcome[3], even with the same $p_\gamma$, $m$, and $t$.

**Up to this point, we have not specified how one would actually learn the function** $F$**.** However, we show this subtly depends on the underlying assumptions with regards to the *in vitro* system in question. By explicitly stating such assumptions, we find a number of novel formulations logically follow. For example, we show assumptions pertaining to how cellular differentiation is induced can alter the loss function, or how differentiating versus non-differentiating cells are similar but somewhat distinct problems, see Figure 2.

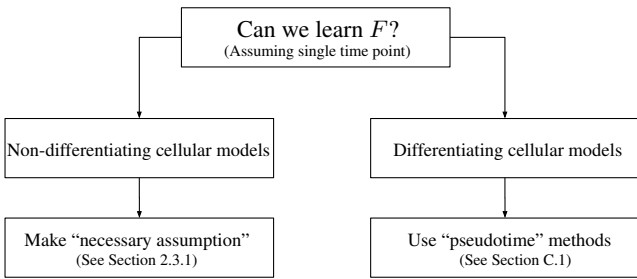

Figure 2: Our ability to learn $F$ depends on the underlying *in vitro* system.

## 2.3 Non-differentiating cellular models

Many *in vitro* cellular models pertain to non-differentiating systems, i.e, left to its own devices, the cells observed after time $t$ is essentially the same as it was at the beginning of the experiment. This observation allows us to make our key necessary simplifying assumption.

In fact, *most* perturb-seq datasets relate to non-differentiating cellular models (Peidli et al., 2024). This is because oftentimes immortalized cancer cell lines are easier to culture, easier to genetically edit, and one does not need to characterize complex cytokine or transcription factor combinations to induce differentiation. In reality, perturb-seq style methods are still an emerging technology and there has been a trend to showcase sequencing methods on simple cell lines before progressing to advanced models.

To construct loss functions, we must first define some notation. We write $\mathbb{G}$ as the set of perturbed genes for $n_P = |\mathbb{G}|$ perturbed genes (or multi-gene perturbations). For each gene $\gamma \in \mathbb{G}$, we write that $p_\gamma \in \mathbb{P}$ for one of the key perturbation types $\mathbb{P} = \{\uparrow, \downarrow, \times\}$. For completeness, we write $\mathbb{P}_0 = \mathbb{P} \cup \{\cdot\}$, where $\cdot$ corresponds to the action of not perturbing the gene in question. The set of all possible perturbation states is then defined as $\mathbb{P}_0^{\mathbb{G}} = \{p_\gamma \in \mathbb{P}_0 | \gamma \in \mathbb{G}\}$.

Analogously, either $m \in \mathbb{M}$, where $\mathbb{M}$ is the set of non-baseline media conditions for $n_M = |\mathbb{M}|$ unique conditions, and $\mathbb{M}_0 = \mathbb{M} \cup \{\cdot\}$ is the total set with the baseline media condition included.

### 2.3.1 Necessary assumption: Unedited cells do not respond to baseline media

Published literature primarily includes experiments that do not characterise their starting material $X$ to the same extent as their typical measured states, see Figure 1A. In the case where unedited cells do not differentiate in the baseline media, we show that our problem simplifies to become tractable using the observations illustrated in Figure 1B. This assumption appears to be implicitly made in many key pieces of work using ML to predict the outcome of genetic perturbations (Roohani et al., 2022). For all $t > 0$, we write

$$F(X, p_\gamma = \cdot, m = \cdot, t) = W(M(P(X, \cdot), \cdot), t) = W(i(i(X)), t)$$
$$= W(X, t) = X_t = X .$$

We also note that softer conditions would likely suffice, e.g., the moments of $X$ and $X_t$ are identical — however we have not defined the space, $\mathcal{X}$, in which $X$ resides. Subsequently, path 4 in Figure 1 is the identity function and measurements of $X_t$ are in fact identically distributed to measurements of

---

[3]If we wanted to flip the order such that media was added before the genomic perturbation (followed by a waiting period), then we would be be trying to learn $F(F(X, \cdot, m, \cdot), p_\gamma, \cdot, t)$.

224  $X$. The function $F$ can then be fit through pairs of input-output data points by mapping $X_t$ in path 4
225  to the end states in paths 1, 2, and 3 in Figure 1.

226  For a loss, $\mathcal{L} : \mathcal{X} \times \mathcal{X} \to \mathbb{R}_+$, applied to predicted-actual pairs $(\hat{X}, X) \in \mathcal{X} \times \mathcal{X}$, we can calculate a
227  total loss $\mathcal{L}_\mathrm{T}$ over all input–output pairs as

$$\mathcal{L}_\mathrm{T} = \underbrace{\sum_{m \in \mathbb{M}} \sum_{\gamma \in \mathbb{G}} \mathcal{L}\left(F(X, \mathrm{p}_\gamma, m, t), X_{m,t}^{\mathrm{p}_\gamma}\right)}_{\text{path 1: } n_\mathrm{P} n_\mathrm{M} \text{ data points}} + \underbrace{\sum_{\gamma \in \mathbb{G}} \mathcal{L}\left(F(X, \mathrm{p}_\gamma, \cdot, t), X_t^{\mathrm{p}_\gamma}\right)}_{\text{path 2: } n_\mathrm{P} \text{ data points}} \tag{5}$$

$$+ \underbrace{\sum_{m \in \mathbb{M}} \mathcal{L}\left(F(X, \cdot, m, t), X_{m,t}\right)}_{\text{path 3: } n_\mathrm{M} \text{ data points}} + \underbrace{\mathcal{L}\left(F(X, \cdot, \cdot, t), X_t\right)}_{\text{path 4: 1 data point}},$$

228  where $n_\mathrm{P}$ is the number perturbations and $n_\mathrm{M}$ is the number of (non-baseline) media conditions.
229  Across paths 1 to 4, we count a total of $(n_\mathrm{P} + 1)(n_\mathrm{M} + 1)$ pairs of data points.

230  In order to learn $F$, assuming we only measure a *single time point* as shown in Figure 1B and we
231  have a non-differentiating cellular model, we *must* make this necessary assumption that unedited
232  cells do not respond in baseline media.

233  If this necessary assumption cannot be made, the function that learns the relationship between
234  paired measurements of $(X_t, X_{m,t}^{\mathrm{p}_\gamma})$ then becomes a counter factual prediction. If we write that
235  $X_t = F(X, \cdot, \cdot, t)$, then by stating the existence of the inverse function, $X = F^{-1}(X_t, \cdot, \cdot, t)$, we
236  can define a counter factual function as

$$C(X_t, \mathrm{p}_\gamma, m) = (F \circ F^{-1})(X_t, \mathrm{p}_\gamma, m) \tag{6}$$
$$= F(F^{-1}(X_t, \cdot, \cdot, t), \mathrm{p}_\gamma, m, t).$$

### 2.3.2  Optional assumption I: Perturbed distributions are attractors of dynamical systems

238  In early systems biology literature employing large systems of ordinary differential equations (ODEs)
239  various steady state assumptions are typically made to simplify downstream analysis (Klipp et al.,
240  2005). Attractors are stable steady states or regions of state space within a dynamical systems that
241  solutions converge towards. If we make the assumption that $F$ determines a dynamical system and
242  $X_{m,t}^{\mathrm{p}_\gamma}$ is a steady state[4] for some $(\mathrm{p}_\gamma, m) \in \mathbb{P}_0^{\mathbb{G}} \times \mathbb{M}_0$ and $s, t \geq 0$ then

$$\frac{\mathrm{d}}{\mathrm{d}t} F(X_{m,s}^{\mathrm{p}_\gamma}, \cdot, \cdot, t) = 0, \tag{7}$$

243  or in the non-infitesimal case

$$F(X_{m,s}^{\mathrm{p}_\gamma}, \cdot, \cdot, t) = X_{m,s+t}^{\mathrm{p}_\gamma} = X_{m,s}^{\mathrm{p}_\gamma}. \tag{8}$$

244  In essence, this means that the duration of our experiment is much longer than the time needed for
245  cells to reach a steady state (for example, in transcriptional space).

246  As a mechanism to incorporate this into a loss function, this then leads to additional terms in our loss
247  function for these regions of state space

$$\underbrace{\sum_{(\gamma, m) \in S} \mathcal{L}\left(F(X_{m,t}^{\mathrm{p}_\gamma}, \cdot, \cdot, s), X_{m,t}^{\mathrm{p}_\gamma}\right)}_{\text{Up to } (n_\mathrm{P} + 1)(n_\mathrm{M} + 1) \text{ data points}}, \tag{9}$$

248  for subset $S \subseteq \mathbb{G} \times \mathbb{M}$ corresponding to perturbations and media conditions where the dynamical
249  system is believed to have relaxed. As a trivial example, in the TGF$\beta$ example explained in the
250  introduction to Section 2.2: within a knockout screen for a non-differentiating cell model, $S$ could
251  include the element $(TGFBR1, TGFB1) \in S$ because the $TGFB1$ cytokine media condition cannot
252  induce a response in $TGFBR1$ knocked out cells and thus the system is at steady state.

---

[4]Note that our random variable, $X$, can still be at steady state in aggregate, even though individual cells still
progress through the cell cycle.

If we have further time series data, we obtain further paired data points along trajectories as the system approaches the steady state. In Section 4, we demonstrate how enforcing steady states in a NODE model of transcription dynamics using Equation (9) leads to rapid convergence when compared to a loss function without using this additional term. We discuss an alternative "softer" version of optional assumption I in Appendix A.

**Experimental recommendations**

From Section 2.3, we proposed a number of assumptions that allow one to better leverage perturbation data:

- *Verification of the necessary assumption through measurement of $X$: unedited cells do not respond to baseline media[5].*
- *Generation of time series data to validate optional assumption I (or optional assumption II, see Appendix A).*

These assumptions will need validating for any experimental system of interest. As these will typically require comparison of mRNA at different time points, we should note the likely requirement of fixation methods or cascading experiment start times (De Jonghe et al., 2024a,b).

# 3   Measurements of cell state using single-cell technology

One challenge when learning $F$ is that we never *actually* measure random variables within $\mathcal{X}$, we typically measure finite-dimensional count vectors as generated by single-cell 'omic technologies. Foundation models typically aggregate large amounts of data originating from many sources. In a perfect world where we have infinitely many perfect measurements of cell state, the mathematical setup presented in Section 2 would suffice. However, with regards to training foundation models: single-cell omics contains numerous complexities that are not found in many other data types. These relate to the modality used, the depth of sequencing and batch effects driven by biological and technical factors. Therefore, the types of model structures that $F$ will incorporate will be limited by the types of measurement technology and experimental design. We now move from an abstract concept of cell state to specific single-cell omic readouts.

## 3.1   Single-cell technology description and resulting learned function

The central dogma of molecular biology states that biological sequential information is transferred from DNA to RNA as it is transcribed, and from RNA to proteins as it is translated. Modern molecular biology has now advanced to the point that single-cell technologies are now able to measure omic modalities relevant to: chromatin accessibility (a DNA and nuclear protein complex) relevant to describing which areas of DNA are being transcribed; mRNA transcript abundance levels relevant to specifying which genes are active; and specific protein levels illustrating which mRNA were translated (De Jonghe et al., 2024a,b). Originally these biomolecules would be measured separately through single-cell Assay for Transposase-Accessible Chromatin using sequencing (scATAC-seq) (Chen et al., 2018b), single-cell Ribonucleic acid sequencing (scRNA-seq), and Cellular Indexing of Transcriptomes and Epitopes by Sequencing (CITE-seq) (Stoeckius et al., 2017). However, some of these modalities can now be measured simultaneously, for example DOGMA-seq (Mimitou et al., 2021) and TEA-seq (Swanson et al., 2021) are able to measure all of the aforementioned biomolecules. For an illustration of how measurements of key biomolecules are transformed into processed data, see Figure 3. Due to the expense of running such advanced assays, any framework that endeavours to capture large aspects of biology will have to be able to handle incomplete data with missing observations.

Returning to our mathematical construction, we can consider omic readouts as functions applied to $X$, which themselves become random variables that can be sampled from to create a finite dimensional vector. Specifically single-cell ATAC-seq, RNA-seq and CITE-seq measurements can be written as

$$\mathbf{x}_{\mathcal{G}} \sim \mathscr{V}_{\mathcal{G}}(X), \quad \mathbf{x}_{\mathcal{T}} \sim \mathscr{V}_{\mathcal{T}}(X), \quad \mathbf{x}_{\mathcal{P}} \sim \mathscr{V}_{\mathcal{P}}(X) \tag{10}$$

---

[5]It is also worth characterising what exactly in the media is driving the response such that only a minimal set of growth-factor components are required.

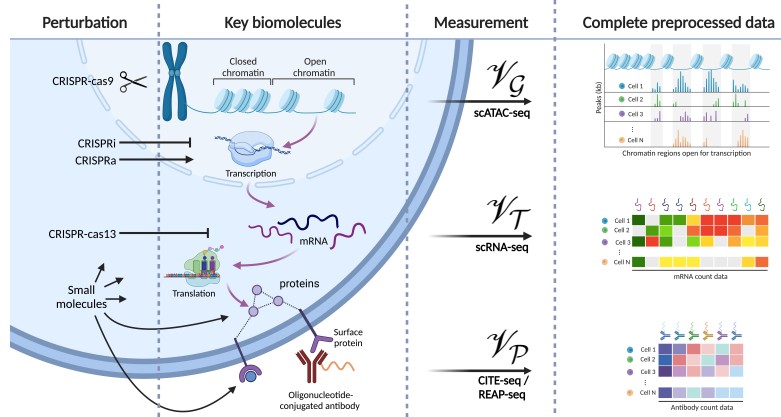

Figure 3: Diagrammatic overview of how an all encompassing variable $X$ is transformed by single-cell technologies into a dataset by $\mathscr{V}$. Adapted from Peidli et al. (2024).

respectfully for $\mathbf{x}_{\mathcal{G}} \in \mathbb{N}_0^{n_{\mathrm{w}} n_{\mathrm{G}}}$ and $\mathbf{x}_{\mathcal{T}}, \mathbf{x}_{\mathcal{P}} \in \mathbb{N}_0^{n_{\mathrm{G}}}$, and we provide a visualisation of these measurements in Figure 3. In Equation (10), we assume that ATAC-seq reads have been binned to $n_{\mathrm{w}}$ windows per gene, and $\mathbb{N}_0$ is used to denote the set of natural numbers including zero. We note that (to date), no assay is able to measure a full state $\mathbf{x}_{\mathrm{TOTAL}} = (\mathbf{x}_{\mathcal{G}}, \mathbf{x}_{\mathcal{T}}, \mathbf{x}_{\mathcal{P}})$, but only a noisy subset of the transcriptome or proteome. To simplify exposition, we will use $\mathscr{V}$ to indicate *some* omic measurement has been made as many of our conclusions are agnostic to measurement technology.

From Equation (4), we can apply $\mathscr{V}$ to both sides to obtain

$$\underbrace{\mathscr{V}(X_{m,t}^{\mathrm{p}_\gamma})}_{\sim \mathbf{x}_{m,t}^{\mathrm{p}_\gamma}} = \mathscr{V}\left(F(X, \mathrm{p}_\gamma, m, t)\right) = \mathscr{V}\left(F(\mathscr{V}^{-1}(\underbrace{\mathscr{V}(X)}_{\sim \mathbf{x}}), \mathrm{p}_\gamma, m, t)\right). \tag{11}$$

Therefore, as we cannot learn $F$, the best we can do is learn the projected function

$$\mathscr{F} := \mathscr{V} \circ F \circ \mathscr{V}^{-1} : \mathbb{X} \times \mathbb{P}_0^{\mathbb{G}} \times \mathbb{M}_0 \times (0, t) \to \mathbb{X}, \tag{12}$$

to the extent that $\mathscr{V}$ has an inverse and $\mathbb{X} = \mathrm{supp}(\mathscr{V}(X))$.

**Key consequences of measuring cell state**

By virtue of measuring the cell state using single-cell technology and learning the projected function $\mathscr{F}$, a few consequences emerge:

- *Batch effects are present by virtue of different sequencing runs, see Appendix B.1). An error analysis leads to further experimental recommendations to minimise these effects, see Appendix B.2.*

- *We need to use specific loss functions to account for the fact that we cannot control how many cells we harvest, see Appendix B.3.*

- *We can build metrics to calculate distances between cells leading to "pseudotime" methods, see Appendix C.1 where we propose use of NODE models.*

## 4  Proof of principle: Optional Assumption I

In Ishikawa et al. (2023), an iPSC model underwent a pooled CRISPR screen with measurements taken on days 2, 3, 4, and 5, but without inducing a terminally differentiated state. In Appendix E, we confirm this from transcriptomic signatures and identify 14 perturbations (including the non-targeting control) that appear to converge on a steady state. For the proof of principle demonstration, we pseudobulk single-cell data over $(\mathrm{p}_\gamma, t)$ pairs giving 100 unique data points. To reduce the number of genes, only those that significantly varied over the time course were selected, leaving $n_{\mathrm{G}} = 120$

For the scenario without any changes to the media, that is, $\mathbb{M} = \emptyset$, then $\mathscr{F} = \mathscr{F}(\mathbf{x}, \mathrm{p}_\gamma, t)$. Using the NODE model in Equation (25), we predict unseen $(\mathrm{p}_\gamma, t)$ pairs at time $t = 5$ for 11 non-steady state perturbations. We train the model using the remaining data via the original loss function given

by Equation (5), or a loss function with steady states enforced using a modification analogous to Equation (9).

We report the test MSE curves in Figure 4 and find that the modified loss function leads to improved performance and stability of the underlying NODE model. Therefore, enforcing steady states in some regions of $\mathbb{X}$ improves predictions of other transcriptional states *not* at steady state by regularizing the overall space of possible functions attainable by the neural network!

## 5 Discussion

From the exposition, we have presented a unified framework that encompasses many of the published ML models developed for single-cell perturbation screens. Moreover, we have uncovered and clarified many hidden assumptions taken for granted by the ML community. By building gold-standard datasets using the experimental recommendations presented, we can systematically identify what assumptions and ML architectures work and which do not. From here, we will be in a position to build foundation models that have a robust capacity for extensive out-of-distribution generalization: to predict transcriptomic states, and phenotypes, for cells modified by novel perturbation in stimulated and unstimulated conditions across time.

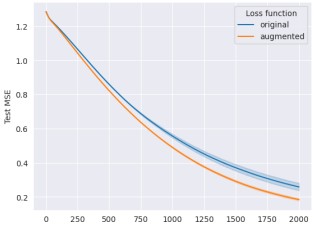

Figure 4: Mean squared error curve on the test set for numerical proof of principle for the modified train loss function in Section 2.3.2. Experiment is repeated 100 times with different random seeds for each loss function.

Although we are a long way away from this vision, we believe that building first-principles approaches is the most promising starting point for such foundation models. There are also substantial routes to strengthening our mathematical formalism: we have not yet considered cell-cell interactions or cell cycle effects — of which will be the subject of future work. Other groups have also proposed mechanisms to combine biophysical modelling with deep learning frameworks (Carilli et al., 2024), suggesting we are not the only group thinking in this manner.

### 5.1 Alternative Views

As an alternative, one may consider building a suitably general ML model and then let the model "figure out the rules" via active learning or reinforcement learning (Scherer et al., 2022; Bertin et al., 2023). Whilst promising, our view is that regardless of the vast datasets now being generated, meaningful progress has yet to be realised as demonstrated by the unreasonable effectiveness of linear models (see introduction).

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

# A  Optional assumption II: Genetic perturbations do not induce responses in baseline media

The steady state assumption detailed in Section 2.3.2 may be too extreme for many complex experimental systems. However, there is an alternative to this assumption that may be more appropriate.

For knock out screens in particular, genetic perturbations often exhibit few differentially expressed genes (DEGs) in the baseline media condition. This is because the cell does not actively require the protein to respond to the nascent signalling cascades triggered by this baseline media condition. For example, it could be that the cell is slowly proliferating and the protein is not involved in cell cycle or background metabolic processes. In contrast, once the cell is stimulated by the addition of a component to the media, the effect can be profound once the cell needs a protein to process the response.

There are a few papers where we observe this effect, including Frangieh et al. (2021); Jiang et al. (2024). To this end, some experimental protocols elect not to use the baseline media condition for knockout screens to save on costs, see Papalexi et al. (2021). For an illustration of this effect, we show a Uniform Manifold Approximation and Projection (UMAP) in Figure 5. Here, we see enhanced effects of CRISPRi perturbations within an iPSC model of astrocytes (Leng et al., 2021) when exposed to a cocktail of IL-1$\alpha$, TNF and C1q cytokines versus a baseline media background condition. In the baseline media condition, both the perturbed cells and non-targeting control cells appear to be drawn from the same distibution; in the stimulated condition the perturbations form clusters. When examining DEGs, regardless of the specific thresholds (log2 fold changes and $p$-values) used to calculate DEGs, we see approximately twice as many DEGs in the stimulated media (i.e., $X_{m,t}^{\mathrm{p}_\gamma}$ vs $X_{m,t}$) when compared to the baseline media condition (i.e., $X_t^{\mathrm{p}_\gamma}$ vs $X_t$).

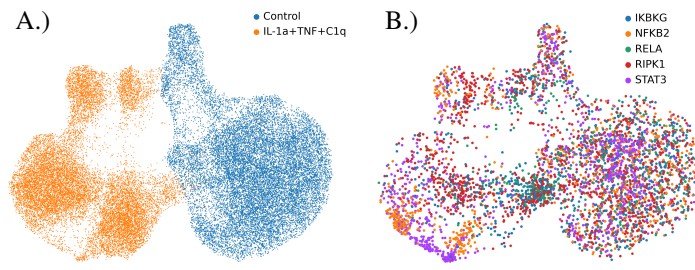

Figure 5: UMAP embeddings of astrocyte perturb-seq data, in (A.) cells are coloured by media condition, and in (B.) select perturbations shown.

In section 2.3.1, we collapsed path 4 from Figure 1 into the identity function. If we make the further assumption that cells are not induced to differentiate by the perturbation in the baseline media, then for some $(\gamma, \cdot) \in S \subseteq \mathbb{G} \times \mathbb{M}$

$$
\begin{aligned}
F(X, \mathrm{p}_\gamma, m = \cdot, t) &= W(M(P(X, \mathrm{p}_\gamma), \cdot), t) \\
&= W(M(X^{\mathrm{p}_\gamma}, \cdot), t) \\
&= W(X^{\mathrm{p}_\gamma}, t) \\
&= X_t^{\mathrm{p}_\gamma} \\
&= X^{\mathrm{p}_\gamma} .
\end{aligned}
\tag{13}
$$

In words, we have collapsed path 2 into the identity function from $*$ in Figure 1 as $X$ becomes time and media invariant. We can then get additional input-output data pairs, by inserting $X^{\mathrm{p}_\gamma}$ at $*$ and predicting outputs $X_{m,t}^{\mathrm{p}_\gamma}$ along path 1. This then generates an additional term within the loss function

$$
\underbrace{\sum_{(\gamma, \cdot) \in S} \mathcal{L}\left( F(X^{\mathrm{p}_\gamma}, \cdot, \cdot, t), X_t^{\mathrm{p}_\gamma} \right)}_{\text{path 1-1: up to } n_\mathrm{P} n_\mathrm{M} \text{ data points}},
\tag{14}
$$

for subset $S \subseteq \mathbb{G}$ of the perturbations where this effect is observed.

# B  Measurements of cell state using single-cell technology (continued)

## B.1  Measurement artifacts

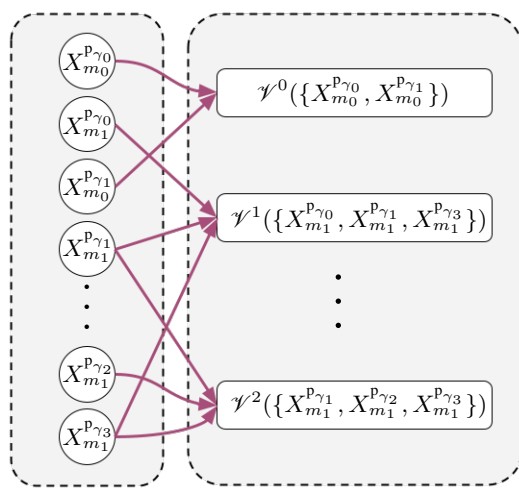

Figure 6: The set of actions in Figure 1 is expanded to include the process of measuring cell state using single-cell technologies. Note that aspects of experimental design can impact how batch effects may emerge; the new arrows and measured states are specific to the description in the text.

Focusing on the most commonly used single-cell omic modality, scRNA-seq, there are two technical caveats that one must consider when modelling the resulting data: dropout and batch effects.

Briefly, dropout refers to the inability for many of the popular single-cell sequencing technologies to detect lowly expressed reads, in fact only $\sim$5-30% of transcripts are actually measured in the cell, and these measurements *may* be biased towards highly expressed genes. Various models have been chosen as the measurement function $\mathscr{V}_{\mathcal{T}}$ to account for this, most commonly via the Zero-Inflated Negative Binominial (ZINB) model [6]. There are similar technical artefacts when considering scATAC-seq data. For reviews pertaining to the modelling of single-cell count data, see (Jiang et al., 2022; Choudhary and Satija, 2022).

Batch effects, or small differences between experimental runs can be much more pernicious and emerge for a number of reasons; these are typically attributed to either biological variation or technical variation. Biological variation corresponds to differences in the cellular model of interest, e.g., an immortalised cell model has been allowed to undergo more rounds of cell division than another cell model that is *supposedly* identical. Technical variation relates to the imperfections in the manufacturing process for biological instrumentation and reagents. Of particular relevance to single-cell technologies: most technologies sample $\sim$10,000 cells at a time[7].

For CRISPR-based genetic screens, cells are typically edited to express the sgRNA constitutively; subject to a few nuances, this means that not only is the target gene edited, but the identity of the genetic perturbation can be resolved from RNA sequencing data. For cell populations maintained in one of several medias of interest, one typically runs each cell population through a separate single-cell reaction, leading to irresolvable batch effects: one cannot be definitively sure that differences in gene expression are driven by differences in media or imperfections between single-cell sequencing reactions. For an illustration of batching, see appendix Figure 6. For this reason, replicates should be performed[8] — but are often not. We briefly discuss how batch effects fit into our model framework.

One can assume that for each batch we largely measure the true gene expression distribution, supplemented by a zero-mean noise term, $\eta^b$, such that when averaged over batches $\mathbb{E}[\eta^b] = n_{\text{B}}^{-1} \sum_{b=0}^{n_{\text{B}}-1} \eta^b = \mathbf{0}$, and we write

$$\underbrace{\mathscr{V}^b(X)}_{\sim \mathbf{x}^b} = \underbrace{\mathscr{V}(X)}_{\sim \mathbf{x}} + \eta^b(\dots) \tag{15}$$

---

[6]Whilst not the focus of this work, Powell et al. (2024) attribute the presence of zero-inflated counts to heterozygosity.

[7]For microfluidic systems, we typically refer to sequencing $\sim$10,000-20,000 cells as using a reaction within a "chip lane". New non-microfluidic technologies are now available with fewer limitations on reaction sizes, but potentially with other technical limitations — see review (De Jonghe et al., 2024a,b).

[8]Note that one can use "hashing" (barcoded antibodies targeting ubiquitously expressed surface proteins) to remove potential batch effects. Here, the antibody barcode is used to encode the identity of the sample (for example, relevant to a media condition, cell model, or donor) in a unique sequence, and thus cell populations are mixed and the identities of the samples can be re-identified later

for $b = 0, \ldots, n_{\mathrm{B}} - 1$. The key issue is that $\eta^b$ is a *function dependent on which cell states and perturbations* are contained within each batch. Therefore, our ability to learn $\mathscr{F}$ in Equation (12) depends on how $X_t$, $X_t^{\mathrm{p}_\gamma}$, $X_{m,t}$ and $X_{m,t}^{\mathrm{p}_\gamma}$ become batched together. For a brief analysis of the consequences of this, see Appendix B.2 where we investigate how errors propagate across batches and interfere with our ability to learn $\mathscr{F}$. Common to foundation models, there are also careful considerations one must make with regards to combining datasets from multiple laboratories, discussed more in Appendix B.2.

## B.2 Error analysis

We want to examine how far the "true" function $\mathscr{F} = \mathscr{F}(\mathbf{x}, \mathrm{p}_\gamma, m, t)$ is away from a learnt function $\mathscr{F}_* = \mathscr{F}_*(\mathbf{x}, \mathrm{p}_\gamma, m, t)$. The true function has access to unbiased measurements of gene expression, whereas the learned function relies on data from batched single-cell sequencing runs; each batch will contain different perturbations and media conditions. To combat this, we typically either preprocess the data to account for batch effects, or use a ML model that accounts for the batch identity. As preprocessing relies on some underlying statistical model, this can lead to the introduction of hidden confounding factors in the now processed dataset. In contract, including the batch identity is cleaner and allows for end-to-end training all of the way through the raw data.

Focusing on incorporating the batch identity into the learned function, we are essentially interested in learning some form of *average* over functions, $\mathscr{F}_L$, which incorporate batch information. To introduce notation, we define the valid set $V$ as

$$V := \left\{ (i, j, b) : \text{Perturbation } \mathrm{p}_{\gamma_i} \text{ in media } m_j \text{ is contained within batch } b \right\} \tag{16}$$

which will allow us to selectively take sums over scenarios when $(\mathrm{p}_{\gamma_i}, m_j)$ is contained within the batch of interest. We can visualise this as a graph, see Figure 6. Without loss of generality and to shorten the notation, we define $\mathrm{p}_{\gamma_i} = \cdot$ when $i = 0$ and $m_j = \cdot$ when $j = 0$. We use $\chi_V(i, j, b)$ as the indicator function to selectively sum over perturbation-media pairs, $(\mathrm{p}_{\gamma_i}, m_j)$, that are contained within batch $b = 0, \ldots, n_{\mathrm{B}}$, and refer to $V_{0,0}$ as the set of batches where unperturbed cells in baseline media, $X$, have been measured.

To enable analytical progress, consider

$$\mathscr{F}_*(\mathbf{x}, \mathrm{p}_\gamma, m, t) = \frac{1}{n_{\mathrm{B}}} \sum_{b=0}^{n_{\mathrm{B}}-1} \frac{1}{|V_{0,0}|} \sum_{b'=0}^{n_{\mathrm{B}}-1} \chi_{V_{0,0}}(b') \mathscr{F}_L(\mathbf{x}, \mathrm{p}_\gamma, m, t, b' \to b), \tag{17}$$

where $\mathscr{F}_L = \mathscr{F}_L(\mathbf{x}, \mathrm{p}_\gamma, m, t, b' \to b)$ is a function that takes in measurement $\mathbf{x}$ made in batch $b'$, applies perturbation-media-time triplet, $(\mathrm{p}_{\gamma_i}, m_j, t)$ to make a prediction $\mathbf{x}_{m_j,t}^{b,\mathrm{p}_{\gamma_i}}$ in batch $b$. The function $\mathscr{F}_L$ satisfies

$$\mathscr{F}_L(\mathbf{x}^{b'}, \mathrm{p}_\gamma, m, t, b' \to b) = \mathbf{x}_{m,t}^{b,\mathrm{p}_\gamma}, \tag{18}$$

whereby a measurement made in batch $b$ is modified by order $\varepsilon$ error function $\eta$, written

$$\mathbf{x}_{m_j,t}^{b,\mathrm{p}_{\gamma_i}} = \mathbf{x}_{m_j,t}^{\mathrm{p}_{\gamma_i}} + \eta^b(\{\mathbf{x}_{m_j,t}^{\mathrm{p}_{\gamma_i}} : (i, j, b) \in V\}). \tag{19}$$

Equation (19) clarifies Equation (15) by highlighting the dependence that the error depends on the set of perturbations and media conditions in the batch. For example, if a perturbation, $\mathrm{p}_\gamma$, stresses a cell then it may become permeable leading to loss in cytoplasmic RNA; this often leads to fewer RNA reads being captured with said reads disproportionately originating from mitochondria.

We would like to examine the error at point $\mathbf{x}$ over all $(\mathrm{p}_\gamma, m, b)$ triplets, written as

$$\varepsilon(\mathbf{x}, t) = \frac{1}{n_{\mathrm{P}} n_{\mathrm{M}}} \sum_{i=0}^{n_{\mathrm{P}}} \sum_{j=0}^{n_{\mathrm{M}}} \left[ \underbrace{\mathscr{F}(\mathbf{x}, \mathrm{p}_{\gamma_i}, m_j, t)}_{= \mathbf{x}_{m_j,t}^{\mathrm{p}_{\gamma_i}}} - \mathscr{F}_*(\mathbf{x}, \mathrm{p}_{\gamma_i}, m_j, t) \right]^2. \tag{20}$$

Using the Taylor expansion of $\mathscr{F}_L$ around $\mathbf{x}$, we find

$$
\begin{aligned}
\mathscr{F}_L(\mathbf{x}, \mathrm{p}_\gamma, m, t, b' \to b) &= \mathscr{F}_L(\mathbf{x}^{b'} - \eta^{b'}, \mathrm{p}_\gamma, m, t, b' \to b) \\
&= \mathscr{F}_L(\mathbf{x}^{b'}, \mathrm{p}_\gamma, m, t, b' \to b) - \eta^{b'} \cdot \nabla \mathscr{F}_L(\mathbf{x}^{b'}, \mathrm{p}_\gamma, m, t, b' \to b) + \ldots \\
&= \mathbf{x}^{b,\mathrm{p}_{\gamma_i}}_{m_j,t} - \eta^{b'} \cdot \nabla \mathscr{F}_L(\mathbf{x}^{b'}, \mathrm{p}_\gamma, m, t, b' \to b) + \ldots \\
&= \mathbf{x}^{\mathrm{p}_{\gamma_i}}_{m_j,t} + \eta^b - \eta^{b'} \cdot \nabla \mathscr{F}_L(\mathbf{x}^{b'}, \mathrm{p}_\gamma, m, t, b' \to b) + \ldots
\end{aligned}
\tag{21}
$$

and therefore

$$
\begin{aligned}
\varepsilon(\mathbf{x}, t) &= \frac{1}{n_{\mathrm{P}} n_{\mathrm{M}}} \sum_{i=0}^{n_{\mathrm{P}}} \sum_{j=0}^{n_{\mathrm{M}}} \left[ \mathscr{F}(\mathbf{x}, \mathrm{p}_{\gamma_i}, m_j, t) - \mathscr{F}_*(\mathbf{x}, \mathrm{p}_{\gamma_i}, m_j, t) \right]^2 \\
&= \frac{1}{n_{\mathrm{P}} n_{\mathrm{M}}} \sum_{i=0}^{n_{\mathrm{P}}} \sum_{j=0}^{n_{\mathrm{M}}} \left[ \mathbf{x}^{\mathrm{p}_{\gamma_i}}_{m_j,t} - \left( \frac{1}{n_{\mathrm{B}}} \sum_{b=0}^{n_{\mathrm{B}}-1} \frac{1}{|V_{0,0}|} \sum_{b'=0}^{n_{\mathrm{B}}-1} \chi_{V_{0,0}}(b') \mathscr{F}_L(\mathbf{x}, \mathrm{p}_\gamma, m, t, b' \to b) \right) \right]^2 \\
&= \frac{1}{n_{\mathrm{P}} n_{\mathrm{M}} n_{\mathrm{B}}} \frac{1}{|V_{0,0}|} \sum_{i=0}^{n_{\mathrm{P}}} \sum_{j=0}^{n_{\mathrm{M}}} \sum_{b=0}^{n_{\mathrm{B}}-1} \sum_{b'=0}^{n_{\mathrm{B}}-1} \chi_{V_{0,0}}(b') \left[ -\eta^b + \eta^{b'} \cdot \nabla \mathscr{F}_L(\mathbf{x}^{b'}, \mathrm{p}_\gamma, m, t, b' \to b) + \ldots \right]^2
\end{aligned}
\tag{22}
$$

Examining the final line of Equation (22), we find some interesting conclusions, namely that:

- Even sequencing all perturbations and all media conditions in the same batch does not strictly mean one can learn $\mathscr{F}$ unless $\nabla \mathscr{F}_L(\mathbf{x}^{b'}, \ldots) \approx \mathbf{1}$.

- For $(n_{\mathrm{B}} > 1)$, as the $\eta^b$ and $\eta^{b'}$ terms have opposite signs, the total error can be reduced by incorporating unperturbed cells in the baseline media into every batch.

- For $(n_{\mathrm{B}} > 1)$, the first term in the square brackets suggests that some batch effects are irreducible, however modelling $\mathscr{F}_L$ via convex Lipschitz functions would be desirable if possible, because

$$
||\nabla \mathscr{F}_L(\mathbf{x}^{b'}, \ldots) - \nabla \mathscr{F}_L(\mathbf{x}, \ldots)|| \leq L ||\mathbf{x}^{b'} - \mathbf{x}||.
$$

**Experimental recommendations:** From the analysis in Appendix B.2, we find that the total error is a function of the error in batches that contain unperturbed cells in the baseline media, $X$, and the batches containing perturbed cells in stimulated media $X^{\mathrm{p}_\gamma}_{m,t}$. Therefore, assuming that the errors from each batch are independent and identically distributed, *the total error can be reduced by incorporating unperturbed cells in the baseline media into every batch*[9].

**Implications for Foundation Models:** Foundation models for regulatory systems typically rely on data gathered by different laboratories, making a deep understanding of batch effects essential. In principle, technical variation is more tractable because its sources can often be pinpointed. For instance, in scRNA-seq workflows, differences in cell isolation methods, reverse transcription efficiencies, and PCR amplification introduce noise and batch effects, as do variations in capture efficiency and library preparation chemistries (e.g., 10x Genomics, Parse, etc.). Addressing these issues requires careful experimental design, appropriate controls, and standardization or batch-correction methods during data analysis. In theory, many of these technical factors might also be modeled with machine learning and probabilistic approaches.

However, biological variation is more difficult to manage. There is no universal standard for cell lines or culture conditions, so it is challenging to obtain consistent signals across multiple systems. Although useful insights can still be gained from heterogeneous data, explicitly accounting for these biological differences often requires simplistic approaches (e.g., one-hot encoding) rather than richer parametric modeling. Together, the compounding effects of technical and biological variation can distort or mask the signal of interest.

Consequently, validating foundation models on wholly independent test datasets — without extensive data harmonization that risks introducing data leakage — should be a top priority. Looking ahead,

---

[9]One would need to achieve this though a barcoding strategy to combine media conditions into the same chip lane.

automation protocols offer a promising route to generate large-scale standardized datasets that can support robust, generalizable models. Yet any approach that integrates data across multiple sources must do so with a clear awareness of how both technical artifacts and unstandardized biology can impede real-world predictive performance.

## B.3 Loss functions

In Section 2.3.1, we referred to a generic loss function $\mathcal{L} : \mathcal{X} \times \mathcal{X} \to \mathbb{R}_+$ with $(\hat{X}, X) \in \mathcal{X} \times \mathcal{X}$ for illustrative purposes. Depending on the omic measurement(s) taken, we will need to define an appropriate loss function.

With single-cell technologies, one does not have control over exactly how many cells one will capture. Therefore, one is left with the challenge of comparing two distributions: the set of model predictions generated from applying $\mathscr{F}$ to $I$ non-targeting cell population in the baseline media, with $J$ actual perturbed cells. Put simply, we need to construct a loss function between two groups of cells with different numbers of cells contained within each group. More specifically,

$$\mathcal{L}_{\mathscr{V}} \left( \left\{ \mathscr{F}(\mathbf{x}_t[i], \mathbf{p}_\gamma, m) \right\}_{i=1}^{I}, \left\{ \mathbf{x}_{t,m}^{\mathbf{p}_\gamma}[j] \right\}_{j=1}^{J} \right), \tag{23}$$

and therefore $\mathcal{L}_{\mathscr{V}} : \mathbb{X}^I \times \mathbb{X}^J \to \mathbb{R}_+$.

With this challenge in mind optimal transport has been of increased interest to the single-cell community (Bunne et al., 2023, 2024), but contains challenges with respect to the curse of dimensionality. Various methods from statistics are also appropriate, for example use of E-distance (Peidli et al., 2024), or simpler techniques including minimising the mean squared error (MSE) between low order moments (i.e., mean, variance etc). Finally, a number of other hueristics have been tried, including random matching of cells between control and perturbed distributions (Roohani et al., 2022).

# C   Other experimental designs

Thus far, we have covered non-differentiating pooled screens in Section 2.3. Now we have an understanding of how single-cell technology measures cell state, we highlight an exciting new area of inquiry: differentiating cellular models, particularly via the use of *in vivo* systems. In Appendix C.2, we briefly discuss arrayed screens and other modelling assumptions worthy of consideration.

## C.1   Differentiating cell models and *in vivo* systems

By optimising the time point at which cells are harvested, one can capture a range of different differentiation states along a trajectory within a single experiment. To achieve such complex behaviour, cells require stimulation by a cocktail of cytokines *in vitro*, or naturally through the use of *in vivo* perturb-seq screens (where media changes are not possible, $\mathbb{M} = \emptyset$). Shown in Figure 7, Lara-Astiaso et al. (2023) demonstrated an *in vivo* perturb-seq screen to investigate the differentiation of hematopoietic stem cells (HSCs) into myeloid, erythroid, and lymphoid lineages within an irradiated mouse model. After 14 days, we find exogenous CRISPR edited cells in the bone marrow. Cells without edits (the non-targeting population) achieve all 3 lineages, but certain lineages no longer develop when specific proteins are knocked out and these edited cells remain in a HSC state.

This is a universal phenomena in such screens: in experiments wherein cells are encouraged to

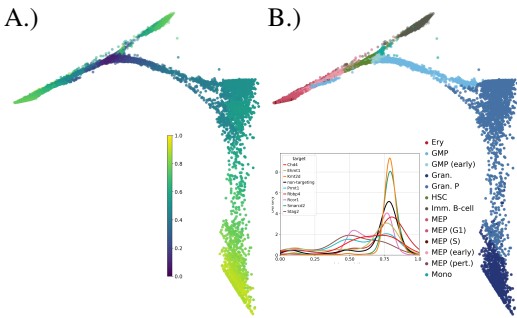

Figure 7: Illustration of haematopoetic stem cells differentiating into myeloid, erythroid, and lymphoid lineages. In panel (A.) we mark each cell with its corresponding pseudotime value, and in (B.) we label each point by estimated cell type. Inset, we see the distribution of different knockout populations along the trajectory.

differentiate, we observe an imperfect process leading to multiple subpopulations and retention of earlier undifferentiated states (Taylor-King et al., 2020b), i.e., there is a (stochastic) drift in the distribution of cell states to include further differentiated cell states. Or in our mathematical notation, for $t > 0$, we find

$$\text{supp}(X_0) \subset \text{supp}(X_t), \tag{24}$$

where $\text{supp}(X) := \{x \in X : p_X(x) > 0\}$ and $p_X(x)$ is the probability density function associated to random variable $X$. In Equation (24), we are specifying that the space of possible states increases over time as some of the cells achieve differentiation into terminal states.

For a single-cell transcriptomic readout with $n_s$ cells passing quality control pipelines, we write $\{\mathbf{x}_i\}_{i=1}^{n_s} \sim \mathscr{V}_{\mathcal{T}}(\{X_t, X_t^{p_\gamma}\})$. Pseudotime methods attempt to derive a mapping $\sigma : \{1, \ldots, n_s\} \to (0, t)$ such that $\{\mathbf{x}_{\sigma(i)}\}$ is ordered in time. In Lara-Astiaso et al. (2023), a pseudotime method was applied to the non-targeting control population only $\{\mathbf{x}_s\} \sim \mathscr{V}_{\mathcal{T}}(X_s)$ to get time labels $s \in (0, t)$. Thereafter, perturbed cell populations $\{\mathbf{x}_s\} \sim \mathscr{V}_{\mathcal{T}}(X_s^{p_\gamma})$ were given a pseudotime value based on their nearest neighbour within the non-targeting control population. Thus, we have a pseudotime value for perturbed and non-targeting cell populations within the dataset.

If we were to learn a function $F$ that maps non-targeting cells with smaller pseudotime values to perturbed cells with larger pseudotime values, we have an opportunity to use modern ML methods with some developments offering a natural framework to approach this phenomena. From Section 3, we approximate $F$ by finite dimensional approximation $\mathscr{F} = \mathscr{V} \circ F \circ \mathscr{V}^{-1}$. In the case whereby there is no branching in the pseudotime process, $\mathbf{x}_s \in \mathbb{X}$ maps a continuous path for $s \in (0, t)$. We can then write $\mathscr{F}$ as the solution to a neural ODE (NODE)[10] (Chen et al., 2018a)

$$\mathscr{F}(\mathbf{x}, p_\gamma, t) = \mathbf{x}_0 + \int_0^t \mathscr{G}(\mathbf{x}, p_\gamma, s) \, \mathrm{d}s, \tag{25}$$

for neural network $\mathscr{G}$. When branching differentiation trajectories occur, natural extensions to NODEs can be employed, e.g., neural stochastic differential equations (Kidger, 2022).

## C.2 Arrayed screens

In contrast to pooled screens, arrayed screens can be used to understand more complex phenotypes whereby cells are interacting with each other and their environment, e.g., bone formation (Taylor-King et al., 2020a). Practically, arrayed screens are both more complicated and simpler than pooled screens for a number of reasons. On one hand, challenges include that each well on a plate (96 well, 384 well plates etc) becomes a batch with the potential for *edge effects* — whereby the outer rim of the plate may have slightly weaker or stronger phenotypes. On the other hand, some phenotypes emerge from cell-cell signalling and can even by triggered by nearby cells; therefore in such set ups the evolution of random variable $X_{m_j}^{p_{\gamma_i}}$ is entirely independent of all other perturbation-media pairs $X_{m_{j'}}^{p_{\gamma_{i'}}}$ with $i \neq i'$ and $j \neq j'$.

# D Connection to other areas of machine learning literature

## D.1 Variational autoencoders

We note that there have been many ML models utilising variational autoencoders (VAEs) to model cellular responses. We show that this is a special case of the mathematical construction presented thus far, by noting the assumption that recovery of $\mathbf{x}_{m,t}^{p_\gamma}$ is only dependent on a latent variable

$$\mathbb{P}\left(\mathbf{x}_{m,t}^{p_\gamma} \mid \mathbf{x}, p_\gamma, m, t\right) = \int \underbrace{\mathbb{P}\left(\mathbf{x}_{m,t}^{p_\gamma} \mid y, \mathbf{x}, p_\gamma, m, t\right)}_{=\mathbb{P}\left(\mathbf{x}_{m,t}^{p_\gamma} \mid y\right)} \mathbb{P}\left(y \mid X, p_\gamma, m, t\right) \mathrm{d}y \tag{26}$$

and the act of $p_\gamma$, $m$, and $t$ act via the function $L$, thus

$$\mathbb{P}\left(y \mid X, p_\gamma, m, t\right) = \int \delta\left(y - L(z, p_\gamma, m, t)\right) \mathbb{P}\left(z \mid \mathbf{x}, p_\gamma, m, t\right) \mathrm{d}z. \tag{27}$$

---

[10]NODEs have recently been employed (Cui et al., 2022) in the development of RNA velocity models (La Manno et al., 2018) — a related but distinct problem.

Therefore, by enforcing $z$ to be normally distributed, we recover the VAE-style formulation.

$$\mathbb{P}\big(\mathbf{x}_{m,t}^{\mathrm{p}_\gamma}|\,\mathbf{x},\mathbf{p}_\gamma,m,t\big) = \int \mathbb{P}\big(\mathbf{x}_{m,t}^{\mathrm{p}_\gamma}|\,L(z,\mathbf{p}_\gamma,m,t)\big)\mathbb{P}\big(z|\,\mathbf{x},\mathbf{p}_\gamma,m,t\big)\mathrm{d}z\,. \tag{28}$$

We note that different VAE-based models have used different omic modalities. For example, in Inecik et al. (2022), transcriptomic ($\mathbf{x}_{\mathcal{T}}$) and proteomic data ($\mathbf{x}_{\mathcal{P}}$) are mapped into the same embedded space; Yang et al. (2021) use a similar architecture but tailored to transcriptomic and imaging data. If you have modalities in the same coordinate system, e.g. transcriptomic and proteomic (gene-based), you can map data into the same latent space in a simple manner. When data lives in different coordinate spaces such as transcripomics and imaging, you have to match distributions in the latent space.

In addition to training on data where predictions are matched to empirical truth, when $L$ is the identity function every data point can also be mapped onto itself using a Kullback–Leibler divergence style loss function, generating additional data points equal to the total number of cells.

## D.2 Causal modelling

A substantial body of work (Sussex et al., 2021; Uhler and Shivashankar, 2022; Lopez et al., 2023; Ke et al., 2023; Lagemann et al., 2023; Mao et al., 2024; Kovačević et al., 2024) has focused on using causally-inspired models to predict the effect of interventions while providing an element of interpretability where the aim may be to learn a causal graph $G = (V, E)$ with vertices $V$ and directed edges $E$. Vertices that are $d$-separated in the graph correspond to conditionally independent variables in the data. The number of potential causal graphs that might explain any set of observations can scale hyperexponentially with $|V|$ — making causal structure learning for transcriptomics, where $|V| = n_{\mathrm{G}} \approx 20,000$, very difficult even with interventional data (Uhler et al., 2013). Recent work in causal modelling has worked towards reconciling differences between the observed unperturbed and perturbed distributions by considering them as stationary diffusions (Lorch et al., 2024).

Nevertheless, causal approaches are conceptually attractive, especially where one is interested in causal mechanisms of disease. For example, given a particular desired healthy cell state and an initial diseased cell state, a causal model of perturbations would help identify a perturbation that would take us from the initial state to the desired state. In this case, learning the full input-output mapping across perturbations is not necessary as we are only interested in a particular outcome (Zhang et al., 2023). Experiments and modelling must go hand-in-hand. Developing models that answer biologically relevant questions rather than performing generic prediction will help narrow the causal hypothesis space.

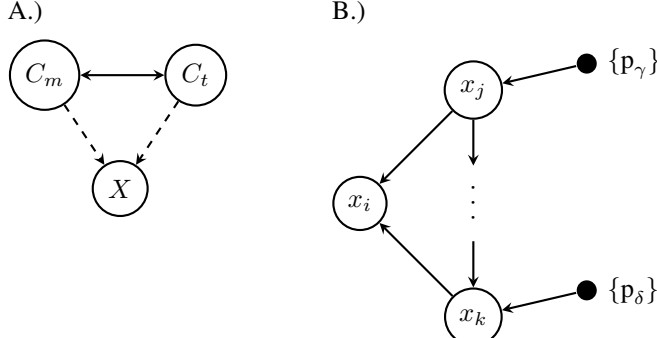

Figure 8: (A.) High level causal graph where each contextual variable potentially acts on all genes within $X$ or some subset. (B.) Gene-level causal graph where $\mathbf{x} = (x_1, \ldots, x_{n_{\mathrm{G}}})$ are gene counts and perturbations (e.g., $\mathrm{p}_\gamma$) are parameters.

Our mathematical framework can be considered as an augmented version of the standard model of causality that has been applied to perturb-seq experiments (Yang et al., 2018; Wang et al., 2017). We treat $\mathrm{p}_\gamma$, $m$ and $t$, as auxiliary context variables or parameters (Magliacane et al., 2016) that act on gene-level elements of the causal structure. In Figure 8(A), $m$ and $t$ are contextual random variables that potentially act on every gene in $X$. Figure 8(B) represents a causal graph on the gene level, where CRISPR perturbations $\mathrm{p}_\gamma$ act on individual genes. Perturbations are parameters instead of

random variables. This adaptation arises naturally as $m$ and $t$ modify cellular context and $p_\gamma$ is a direct intervention on gene expression.

This is one of a number of possible approaches to adapting a causal model to our framework. However, it has been shown in previous work that without taking into account the relevant contextual variables, it is impossible to distinguish certain causal relationships (Mooij et al., 2020).

# E    Analysis of iPSC time series dataset

To assess whether the iPSC cells in Ishikawa et al. (2023) reach steady state we evaluate whether the mean log2 fold change (LFC) decreases across days. We use a baseline computed using the non-targeting guides to evaluate baseline variation in the dataset. In Figure 9, the mean LFC for each guide is shown in blue where each point represents a sequential day comparison. Given that we have samples from days 2, 3, 4 and 5, the blue lines show the mean LFC for comparisons between days 2 and 3, 3 and 4, and 4 and 5. The baseline LFC is obtained by randomly splitting cells with the non-targeting guide for each day into two groups and computing the LFC between the two splits. This is repeated five times to obtain the final baseline.

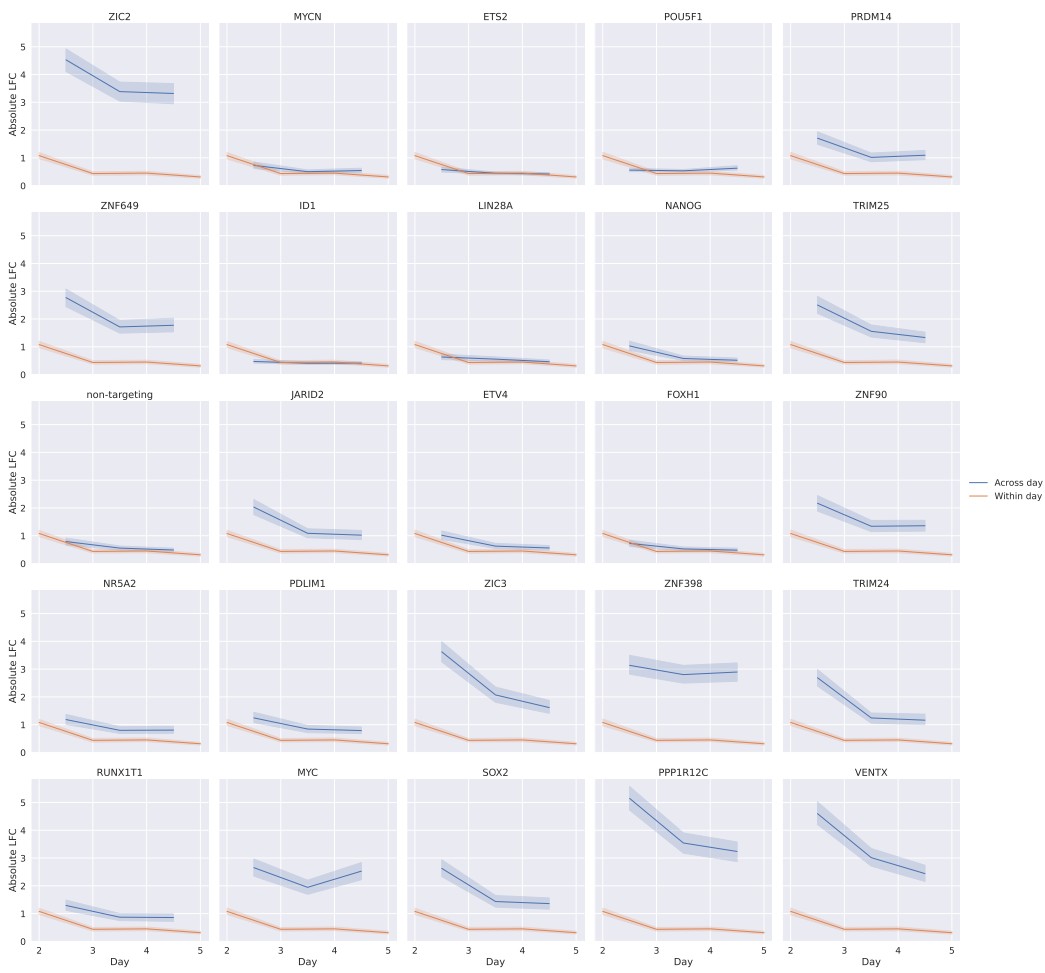

Figure 9: Absolute LFC between sequential days in the Ishikawa et al. (2023) dataset.

