# OpenReview forum: "No Foundations without Foundations: Why semi-mechanistic models are essential for regulatory biology"
_NeurIPS.cc/2025/Position_Paper_Track — Submitted to NeurIPS 2025 Position Paper Track_

### Official Review · Reviewer_Xuwj · 2025-08-03

**Significance:** 3
**Presentation:** 2
**Rating:** 7
**Confidence:** 4

**Summary:**

The paper contends that efforts to build “foundation models” for gene-regulatory biology will fail unless they incorporate mechanistic knowledge rather than rely solely on ever-larger, purely data-driven architectures. It formalises this stance with a semi-mechanistic causal framework that links three experimental regimes—baseline growth, single-gene CRISPR perturbations, and combinatorial interventions—and derives modified loss terms that respect steady-state constraints and unperturbed-cell baselines. The authors show, on a public perturb-seq time-series dataset, that adding their mechanistic loss to a Neural ODE lowers test MSE and yields more stable parameter convergence than a vanilla end-to-end model. They discuss hidden assumptions in current practice (e.g., “cells don’t respond to media” and “perturbations map to unique attractors”) and propose concrete experimental-design guidelines, such as including untreated controls in every batch. The paper closes with a call for coordinated data collection and hybrid loss functions as the only plausible route to reliable regulatory-biology foundation models.

**Strengths:**

Clear, bold stance: “No foundations without foundations” crystallises why blindly scaling data-driven models is insufficient for regulatory biology.

Provides a unified mathematical framework that ties perturb-seq, batch effects, and dynamical systems into a single causal graph, exposing hidden assumptions in prior ML work.

Offers actionable items—modified loss terms, advice to include baseline controls per batch—and demonstrates a proof-of-principle Neural-ODE that improves test MSE on a real CRISPR dataset.

Thorough literature bridge: spans structural biology, causal ML, optimal-transport, and single-cell perturbation studies, positioning the work squarely in ongoing debates.

**Weaknesses:**

Empirical evidence is narrow (one dataset, one toy model); no cross-lab replication or other modalities (e.g., imaging, proteomics) tested.

Core assumptions—cells at baseline steady state, unique attractors—may not hold in primary or differentiating tissues, but diagnostic guidance is minimal.

Alternative data-efficient paths (e.g., active-learning exploration or self-supervised pre-training on multimodal atlases) are acknowledged yet not quantitatively compared.

**Questions:**

Have you tried the mechanistic loss on other modalities (imaging-based pooled screens or RNA + protein multi-omics) to test generality?

How would you validate the “no baseline response” assumption in heterogeneous primary tissues where steady state is unclear?

With limited resources, which lever yields the biggest benefit first: richer perturb-time series, improved batch design, or mechanistic loss terms—and why?

**Alternative Position:**

Yes, and alternative positions are well-considered and addressed by the argument

**Author Identification:**

No.

**Context:**

3

**Discussion:**

3

**Ethics:**

["NO or VERY MINOR ethics concerns only"]

**Position:**

Yes, the paper argues for or against a position related to machine learning.

**Support:**

3

**Thoroughness:**

4

---

### Official Review · Reviewer_vgw8 · 2025-08-08

**Significance:** 3
**Presentation:** 4
**Rating:** 7
**Confidence:** 4

**Summary:**

This paper critically examines the limitations of deep learning approaches in predicting gene expression profiles and introduces a novel semi-mechanistic modeling framework that unifies data from both in vitro and in vivo CRISPR screens. It reveals key implicit assumptions inherent in mainstream prediction methods, leading to the derivation of improved, more biologically grounded loss functions and optimized batching strategies. By explicitly integrating mechanistic insights with data-driven learning, this hybrid framework aims to more accurately capture the complexity of gene regulation. Furthermore, it provides a stronger theoretical foundation for building more reliable and biologically interpretable models of gene expression.

**Strengths:**

1.The paper introduces a novel semi-mechanistic framework that unifies mathematical formalisms for both in vitro and in vivo CRISPR perturbation screens across differentiating and non-differentiating cellular systems, addressing a critical gap in regulatory biology modeling.

2.It successfully diagnoses overlooked assumptions in existing machine learning methods, such as the implicit supposition that unedited cells exhibit no response to baseline media in non-differentiating models, thereby promoting algorithmic transparency.

3.The work delivers practical contributions, including a modified loss function enforcing steady-state dynamics and data-batching strategies grounded in error analysis, which demonstrably improve performance in neural ODE experiments.

**Weaknesses:**

1.The framework's applicability to in vivo systems remains inadequately substantiated, as validation relies solely on in vitro iPSC data despite claims of generalizability to physiological contexts.

2.Counterarguments advocating data-centric approaches are oversimplified, with insufficient engagement against paradigm-shifting successes like AlphaFold, which achieved breakthroughs without explicit mechanistic integration.

3.Scalability concerns arise from increased model complexity due to mechanistic components, yet the paper omits analysis of computational trade-offs or real-world deployment feasibility.

**Questions:**

1.Given the framework's assumption of instantaneous genetic perturbations, could the authors quantitatively evaluate how temporal delays in in vivo CRISPR delivery might compromise prediction accuracy in physiological environments?

2.In scenarios where biological mechanisms remain incompletely characterized, such as non-coding RNA regulation or epigenetic feedback loops, how does the semi-mechanistic approach prevent reverting to de facto black-box modeling while preserving its interpretability claims?

3.To translate experimental recommendations like time-series validation into practice, what specific incentives, collaborative frameworks, or data-sharing standards would the authors propose to align experimental biologists priorities with this methodology ’s requirements?

**Alternative Position:**

Yes, and alternative positions are trivial straw-man arguments

**Author Identification:**

No.

**Context:**

3

**Discussion:**

4

**Ethics:**

["NO or VERY MINOR ethics concerns only"]

**Position:**

Yes, the paper argues for or against a position related to machine learning.

**Support:**

4

**Thoroughness:**

3

---

### Official Review · Reviewer_uY9B · 2025-08-28

**Significance:** 2
**Presentation:** 3
**Rating:** 4
**Confidence:** 2

**Summary:**

This paper presents the position that semi-mechanistic models are essential for regulatory biology. They lay out a strong theoretical framework detailing why this is the case, including mathematical formulations of their assumptions.

**Strengths:**

(S1) The paper presents a clear, logical argument for their position. I especially appreciate the approach of systematically explaining why scaling existing approaches are unlikely to succeed.

(S2) The paper makes an important contribution by explicitly articulating assumptions that are often hidden in ML models for biology. This systematic exposition of experimental design considerations (batch effects, measurement artifacts, etc.) is valuable.

(S3) The paper is well-written and easy to follow. References are made where appropriate, and appear to be consistent with the statements they are supporting.

**Weaknesses:**

(W1) This paper almost reads like a submission that was intended as a technical submission and not as a position paper. It does a better job of describing a proposed regulatory biology foundation model than of presenting a forward-looking vision for a broader field/community.

(W2) The significance of the position within the broader landscape of ML is not elaborated on, which limits the potential for discussion outside the specific field of regulatory biology, and the specific applications discussed.

(W3) There is no discussion of societal significance or importance. Some inclusion of the intersection between the proposed position and societal impact would strengthen the work as a position paper as opposed to a modeling methodology paper.

**Questions:**

(Q1) What broader lessons does your framework offer for ML applied to other complex biological systems beyond regulatory biology?
(Q2) Given that you've identified why current approaches fail, what concrete next steps should the community take?

**Alternative Position:**

Yes, and alternative positions are well-considered and named but not addressed

**Author Identification:**

No.

**Context:**

4

**Details Of Ethics Concerns:**

I have no ethical concerns with the work.

**Discussion:**

2

**Ethics:**

["NO or VERY MINOR ethics concerns only"]

**Position:**

Yes, the paper argues for or against a position related to machine learning.

**Support:**

3

**Thoroughness:**

3

---

### Note · Authors · 2025-08-22

**1-10 Additional Comments:**

No other comments.

**1-11 Submit Again:**

Probably yes

**1-1 Submission Process:**

5

**1-2 Next Year:**

It would be great to hear from researchers outside of AI/ML to discuss the big problems that could be solved with the right research effort.

**1-3 Future Development:**

See above.

**1-4 Interest:**

["Panel discussions with other position paper authors", "Structured debates on controversial topics", "Mentorship programs for early-career researchers"]

**1-5 Thoughtful:**

8

**1-6 Supportive:**

8

**1-7 Technical Aspects Versus Position:**

9

**1-8 Gate Keeping:**

8

**1-9 Camera Ready Changes:**

1. Inclusion of additional clarifying footnotes on perturbation timescales, in response to reviewer vgw8
2. Greater discussion of active learning and self-supervised pretraining in the Alternative Views section as suggested by reviewer Xuwj.
3. Greater discussion of lab automation/standardization of the data generation process motivated by response to reviewer vgw8.

**3-1 Review Response1:**

vgw8

**3-2 Reaction To Review1:**

Thank you for your review of our manuscript, which we will use to improve our work. Our responses to weaknesses (W)/questions (Q) are below.

W1. See also Q1 below. Note that we did not make claims about generalisability to physiological systems — our position is that semi-mechanistic models increase the likelihood of OOD generalisation. There are no datasets showing perturbations paired with single-cell readouts across a range of experimental systems of increasing complexity — so it is challenging for someone to actually make this claim.

W2. AlphaFold should not be an appropriate comparison for two reasons.
A. The AlphaFold architecture does take into account multiple sequence alignments, so the architecture does mimic key aspects of the underlying biology, i.e., evolutionary constraints.
B. AlphaFold concerns structural biology, but our position paper concerns regulatory biology.

W3. The additional loss functions introduced are linear in the number of perturbations/stimuli introduced, so shouldn’t correspond to a meaningfully larger compute requirement.

Q1. Great question. When we refer to “instantaneous” genetic perturbations, we mean: the timescale for genetic editing is irrelevant when compared to the timescale of the cellular model. Alternatively, one could consider PKPD style models to model the editing rate, but this rate would likely need to be determined experimentally. We will mention this in any final manuscripts.

Q2. We did nog claim we could extract interpretable insights. Really our claim is about improving performance by incorporating realistic constraints on the model a priori (whether black box or not) whilst also considering experimental design.

Q3. Whilst compute has been commoditised by AWS, we need similar initiatives to democratize biodata generation. Incentivising this is hard, but companies that make platform technologies would be a natural starting point, e.g., ThermoFisher, 10x Genomics, Illumina etc.

**3-3 Review Response2:**

Xuwj

**3-4 Reaction To Review2:**

Thank you for your review of our manuscript, which we will use to improve our work. Our responses to weaknesses (W)/questions (Q) are below.

W1. This is a great point, but unfortunately there was only one time series perturbational RNA-seq dataset that we identified. Imaging would not be appropriate as we do not have a grounded measurement of “closeness” between two groups of cells. In contrast, with RNA-seq we have the concept of differentially expressed genes. Proteomics would have been great if you had any suggestions for relevant datasets?

W2. Our core assumptions may or may not be correct depending on the specific context in which they are applied, but they are at least testable.

W3. We would view active learning as a separate problem (the model has to be specified before AL is possible) but worthy of greater discussion. In the original manuscript, we were limited by page counts (and time constraints) but we will aim to incorporate this to a greater extent in the alternative views section.

Q1. See above re. metrics for imaging data. However, multiomics would be worthy of exploration if the datasets existed.

Q2. Very few primary tissues ever truly achieve a genuine steady state; perhaps one could argue that this is the case for senescent cells but this is an exception not the rule. The assumption in Section 2.3.1 “Necessary assumption: Unedited cells do not respond to baseline media” is only relevant to “Non-differentiating cellular models” under the broader Section 2.3 heading. We will make this clearer in a future version of the manuscript.

Q3. Taking an educated guess, I would imagine richer time series data with perturbations vs batch design. Depending on the single-cell technology, batch effects are becoming less relevant as learn the drivers of regulation. However, we really need more benchmarking datasets of this type in the public domain to really know for certain.

**3-5 Review Response3:**

uY9B

**3-6 Reaction To Review3:**

Thank you for your review of our manuscript, which we will use to improve our work. Our responses to weaknesses (W)/questions (Q) are below.

W1. All of the mathematical set up we propose is architecture agnostic, but it does require the detailed mathematical set up to nail the point.

W2. Our position specifically concerns regulatory biology, we do not wish to make assertions outside of this field. From the position track paper call: “The goal of this track is to highlight papers that stimulate (productive, civil) discussion on timely topics that need our community’s input”, which doesn't suggest the need for broad positions relevant to the whole field.

W3. We will include this in any final manuscript, but were somewhat word limited in the original manuscript.

Q1. This is a great question. There may be a case that mechanistic-inspired ML architectures and loss functions are appropriate outside of regulatory biology, e.g., in climate modelling, but we know these application areas less well. There may be reasons that this is a bad idea, e.g., physical laws less well understood and abundance of data obfuscate the need to do so. We will endeavour to include such a reflection in any final manuscript.

Q2. Whilst compute has been commoditised by AWS, we need similar initiatives to democratize biodata generation. Incentivising this is hard, but companies that make platform technologies would be a natural starting point, e.g., ThermoFisher, 10x Genomics, Illumina etc.

---

### Meta-Review · Area_Chair_yD5L · 2025-09-20

**Rating:** 7
**Confidence:** 4

**Strengths:**

According to the reviews, the main strengths of this paper are the following:

The paper presents a clear, logical, and strong argument that purely data-driven models are insufficient for regulatory biology and that it is crucial to incorporate mechanistic knowledge.

The authors introduce a novel semi-mechanistic framework to unify mathematical formalisms for different experimental conditions (e.g., in vitro and in vivo CRISPR screens) and connect them within a single causal graph.

A key contribution is to explicitly articulate and diagnose overlooked assumptions common in ML models for biology, which promotes algorithmic transparency.

The work provides practical, actionable items, such as a modified loss function and data-batching strategies, which demonstrably improve model performance on a real dataset.

The paper is well-written, easy to follow, and thoroughly researched, effectively bridging concepts from multiple scientific fields.

**Weaknesses:**

The main weaknesses pointed out by the reviewers are the following:

The proposed framework was only tested on a single dataset and a single model type. There's no cross-lab replication or testing on other data types like imaging or proteomics.

The paper fails to adequately address major successes of purely data-driven models, such as AlphaFold, which succeeded without the mechanistic integration the paper advocates for.

The paper increases model complexity but doesn't analyze the computational trade-offs or the feasibility of real-world deployment. It also fails to quantitatively compare its approach to other data-efficient alternatives like active learning.

Key assumptions, like cells being at a baseline steady state, may not hold true in all biological contexts, and the paper offers little guidance for when these assumptions are violated.

The paper reads more like a technical description of a specific model rather than a broad, forward-looking position paper for the field. It fails to elaborate on the work's significance for the broader machine learning community or its societal impact.

It should be noted that the authors have answered (in my view quite adequately) to most of these comments.

**Questions:**

No further questions beyond those already raised by the reviewers. It should be noted that the authors have answered (in my view quite adequately) to most of the reviewers' questions.

**Ethics:**

None.

**Thoroughness:**

3

---

### Decision · Program_Chairs · 2025-09-26

Reject